# Expression of the E5 Oncoprotein of HPV16 Impacts on the Molecular Profiles of EMT-Related and Differentiation Genes in Ectocervical Low-Grade Lesions

**DOI:** 10.3390/ijms22126534

**Published:** 2021-06-18

**Authors:** Danilo Ranieri, Deborah French, Salvatore Raffa, Luisa Guttieri, Maria Rosaria Torrisi, Francesca Belleudi

**Affiliations:** 1Department of Clinical and Molecular Medicine, Sapienza University of Rome, 00161 Rome, Italy; danilo.ranieri@uniroma1.it (D.R.); deborah.french@uniroma1.it (D.F.); salvatore.raffa@uniroma1.it (S.R.); luisa.guttieri@uniroma1.it (L.G.); mara.torrisi@uniroma1.it (M.R.T.); 2S. Andrea University Hospital, 00161 Rome, Italy

**Keywords:** FGFR2c, epithelial–mesenchymal transition, HPV16, 16E5

## Abstract

Infection with human papillomavirus type 16 (HPV16) is one of the major risk factors for the development of cervical cancer. Our previous studies have demonstrated the involvement of the early oncoprotein E5 of HPV16 (16E5) in the altered isoform switch of fibroblast growth factor receptor 2 (FGFR2) and the consequent expression in human keratinocytes of the mesenchymal FGFR2c isoform, whose aberrant signaling leads to EMT, invasiveness, and dysregulated differentiation. Here, we aimed to establish the possible direct link between these pathological features or the appearance of FGFR2c and the expression of 16E5 in low-grade squamous intraepithelial lesions (LSILs). Molecular analysis showed that the FGFR2c expression displayed a statistically significant positive correlation with that of the viral oncoprotein, whereas the expression values of the epithelial FGR2b variant, as well as those of the differentiation markers keratin 10 (K10), loricrin (LOR) and involucrin (INV), were inversely linked to the 16E5 expression. In contrast, the expression of EMT-related transcription factors Snail1 and ZEB1 overlapped with that of 16E5, becoming a statistically significant positive correlation in the case of Snail2. Parallel analysis performed in human cervical LSIL-derived W12 cells, containing episomal HPV16, revealed that the depletion of 16E5 by siRNA was able to counteract these molecular events, proving to represent an effective strategy to identify the specific role of this viral oncoprotein in determining LSIL oncogenic and more aggressive profiles. Overall, coupling in vitro approaches to the molecular transcript analysis in ectocervical early lesions could significantly contribute to the characterization of specific gene expression profiles prognostic for those LSILs with a greater probability of direct neoplastic progression.

## 1. Introduction

Persistence of infection with high-risk genotypes of human papillomaviruses (HPVs), such as HPV16, is the main factor driving cervical dysplasia and is known to be a major risk factor for cervical cancer development and progression [1]. Twelve HPVs (16, 18, 31, 33, 35, 39, 45, 51, 52, 56, 58, 59) have been widely recognized by the World Health Organisation (WHO) [2] as high-risk cancer-causing types, with additional HPVs (68, 73) as ‘possibly’ cancer-causing types. Among cervical diseases, low-grade squamous intraepithelial lesions (LSILs) mainly preserve the HPV16 episomal genome, whereas high-grade lesions (HSILs) are generally associated with viral genome integration and cancer progression, which involves increased expression of the early HPV16 oncoproteins E6 (16E6) and E7 (16E7) and loss of the oncoprotein E5 (16E5) [2].

Interestingly, even if 16E6 and 16E7 are considered the two main viral oncogenes, results obtained in the in vitro model of episome-associated cervical neoplasia revealed similarities between the episome- and integrant-associated neoplastic progression, which include impairments of differentiation, induction of epithelial-to-mesenchymal transition (EMT), and [3,4] suggesting a possibly more incisive role of other HPV16 early proteins, such as 16E5, in cervical tumorigenesis [3,5]. More recently, it has been proposed that, when expressed in the early stage of carcinogenesis, 16E5 can be a good target for immunotherapy [6].

It is well-known that the 16E5 oncoprotein cooperates with E6 and E7 oncogenes and transforms epithelial cells by deregulating cell growth, survival and differentiation through the modulation of RTKs and their signaling [7,8,9]. We have previously demonstrated that the expression of 16E5 alone is able to perturb epidermal homeostasis, driving fibroblast growth factor receptor 2 (FGFR2) isoform switch, which results in the downregulation of epithelial FGFR2b [10], a receptor involved in keratinocyte differentiation [11], and consequent aberrant expression of the mesenchymal FGFR2c variant, which leads to the induction of EMT and tumorigenic behavior [12,13], as well as impairment of differentiation [14]. The role of 16E5 in the induction of EMT is also confirmed by the more recent finding of its ability to upregulate Met, another receptor tyrosine kinase (RTK) which contributes to EMT and invasive behavior in keratinocytes [15]. Concerning the impact of 16E5 on epithelial cell differentiation, Wasson and coworkers demonstrated that the corresponding E5 protein of HPV18 possibly works in a similar way to 16E5, interfering with the differentiation program of keratinocytes via the downregulation of FGFR2b [16].

In light of these observations, in the present study we aimed to further clarify the role played by 16E5 in early steps of cervical carcinogenesis, investigating the possible link between its expression and altered FGFR2 isoform switch, the induction of EMT-related transcription factors, and the repression of differentiation markers in human LSILs, as well as in the in vitro model of cervical W12 cells, containing episomal HPV16, in which the expression of 16E5 was specifically repressed by siRNA approaches. 

## 2. Results and Discussion

We have previously demonstrated that the expression of 16E5 in epidermal keratinocytes drives FGFR2 isoform switching from epithelial FGFR2b to mesenchymal FGFR2c isoforms [12], and that this event results from the downregulation of epithelial splicing regulatory protein 1 (ESRP1) [12]. Therefore, to investigate if a link between this altered FGFR2 splicing and 16E5 mRNA expression does exist in the in vivo context of the HPV16-positive ectocervical lesions, we first analyzed 16E5, FGFR2b, FGFR2c and ESRP1 mRNA levels in 17 LSIL samples (*p*#) by real-time PCR. The 16E5 transcripts were normalized with respect to their levels in W12 cells cultured at passage 6 (W12p6), in which about 100–200 copies of the E5-expressing HPV episomes per cell were maintained [17]. Five cervical HPV16 negative samples were used as negative controls (C#A-C#E). The results showed that, as previously observed by us [18], LSIL samples displayed a wide variability in 16E5 transcript expression (Figure 1A), which could be due to a high heterogeneity of episomal/integrated HPV16 distribution in these lesions [19]. The FGFR2b and FGFR2c transcript levels were also highly variable (Figure 1B,C); however, whereas FGFR2b appeared inversely linked to 16E5 (Figure 1B), FGFR2c showed a statistically significant positive correlation with 16E5 expression (Figure 1C). Interestingly, ESRP1 mRNA levels appeared homogenously downregulated in all LSIL samples, independently from the 16E5 expression levels (Figure 1D). As expected, compared to HPV16-positive samples, all negative controls displayed a homogeneous and higher expression of FGFR2b and ESRP1 (Figure 1B,D), as well as a reduced expression of FGFR2c transcripts (Figure 1C) compared to the lesional samples. Therefore, the molecular analysis performed on the pathological samples appears to confirm our previous results obtained in the vitro cell models.

To unequivocally demonstrate the direct role exerted by 16E5 on the FGFR2 isoform switching, we shifted our investigations to the in vitro model of W12p6 cells, previously used by us to obtain forced overexpression of the viral oncoprotein [12] but utilized here to analyze the effects of the 16E5-specific depletion by small interfering RNA approaches. Cells were transfected with siRNA for 16E5 (E5 siRNA) or control siRNA, and E5 gene silencing at the transcript level was estimated by real-time PCR (Figure 2A). The results showed that the depletion of 16E5 led to a significant recovery of FGFR2b and ESRP1 (Figure 2A), as well as to inhibition of FGFR2c expression (Figure 2A), showing the key role played by the viral oncoprotein on ESRP1 modulation and consequent FGFR2 altered splicing. The expression levels of FGFR2 isoforms and ESRP1 transcripts in human epidermal keratinocyte HaCaT cell line and in primary human dermal fibroblasts (Hfs) were estimated as controls (Figure 2A).

We have previously reported that the altered FGFR2 splicing induced by 16E5 in epidermal keratinocytes promotes EMT [12], and that the aberrant expression of FGFR2c in these cells is responsible for induction of the EMT-related transcription factors Snail1 and ZEB1 [13]. Moreover, Snail2 has recently been identified as an additional transcription factor contributing to EMT during cervical carcinogenesis [4]. Therefore, to assess the possible link between 16E5 expression and the induction of transcription factors orchestrating upstream of the pathological EMT in the context of the ectocervical epithelium, the expression levels of Snail1, Snail2 and ZEB1 were investigated in W12p6 cells by molecular analysis, which highlighted their slight, but significant, repression in response to 16E5 depletion (Figure 2B). In addition, we have previously shown that 16E5 is also able to counteract epidermal keratinocyte differentiation [10], and that the aberrant expression of FGFR2c leads to defective keratinocyte differentiation and stratification [14]. In light of these observations, we analyzed the possible direct involvement of 16E5 in the impairment of the differentiation program in the context of ectocervical keratinocytes and as a consequence of FGFR2c appearance. The results showed that, in W12p6 cells, the mRNA expression levels of either the early differentiation marker keratin 10 (K10) and the intermediated differentiation marker involucrin (INV) were significantly increased by 16E5 depletion (Figure 2C). Thus, the in vitro model of W12p6 proved to be an effective strategy to highlight the direct link between 16E5 expression and the establishment of pathological events occurring during ectocervical carcinogenesis, such as EMT induction and the impairment of differentiation.

In order to check if this link was also evident in the in vivo context of precocious ectocervical lesions, we first analyzed the expression of EMT-related transcription factors in our cohort of LSILs and found that the transcription factors analyzed displayed an extremely high variability (Figure 3A–C). However, their trend overlapped with that of 16E5 expression (see Figure 1A) and displayed a statistically significant positive correlation in the case of Snail2 (Figure 3B). As expected, in all 16E5-negative controls (C#A-C#E), Snail1, Snail2 and ZEB1 transcripts showed a more homogeneous and lower expression compared to 16E5-positive samples (Figure 3A–C). Finally, we analyzed the expression trend of keratinocyte differentiation markers, and we found that the LSIL samples displayed a wide variability of K10 and INV mRNA levels (Figure 4A,B), although this appeared to be mainly inversely linked with that of 16E5 (see Figure 1A). It has been demonstrated that the late differentiation marker loricrin (LOR) is expressed in differentiated ectocervical cells [20]; therefore, the mRNA levels of this marker were also estimated in our LSIL cohort, showing a similar trend of K10 and INV (Figure 4C). As expected, compared to LSIL samples, all negative controls displayed a higher expression than all the differentiation markers analyzed (Figure 4A–C). 

Overall, our results suggest that approaches of gene silencing applied in in vitro models, such as that of W12p6 cells containing the full-length HPV16 episomal genome and expressing the viral oncoprotein 16E5, proved to be a useful tool to highlight the role of 16E5 in pathological events possibly occurring in episome-associated cervical neoplasia. In fact, using this model, here we demonstrated, for the first time, that the induction of transcription factors governing the upstream induction of EMT, as well as the simultaneous impairment of the differentiation program in the ectocervical context, could be ascribed to 16E5 expression. In addition, our present data indicate that the use of in vitro approaches, when coupled with molecular transcript analysis in ectocervical early lesions, could help in the identification of specific gene expression profiles prognostic for those LSILs with the highest probability of direct progression towards cervical cancer.

## 3. Materials and Methods

### 3.1. Cytological Samples

Cytological samples were obtained after written informed consent from patients attending the Gynecology Day-Clinic of Sant’Andrea University Hospital, Rome. All samples were HPV16-positive at the genotyping test (INNOLipa HPV Genotyping kit, Fujirebio, Gent, Belgium), and with a cytology consistent with low-grade squamous intra-epithelial lesions (LSIL: *n* = 17, *p*#), in accordance with the Bethesda classification system 2001 [21]. HPV-negative cervical samples (*n* = 5, C#A-E) at the genotyping test, assessed as above, were used as negative controls.

### 3.2. Cells and Treatments

The human keratinocyte cell line HaCaT [22], was cultured in Dulbecco’s modified Eagle’s medium, supplemented with 10% fetal bovine serum (FBS) plus antibiotics.

The human cervical keratinocyte cell line W12 originated from a low-grade cervical lesion [17], which retained ~100 to 200 copies of the HPV16 episomes per cell [23]; it was cultured as previously described [17] and used at passage 6 (W12p6). 

Primary cultures of human fibroblasts (HFs) derived from healthy skin were obtained from patients attending the Dermatology Unit of the Sant’Andrea Hospital of Rome; all patients were extensively informed and their consent for the investigation was given and collected in written form, in accordance with guidelines approved by the management of the Sant’Andrea Hospital. Primary cells were isolated and cultured as previously described [24].

For RNA interference and 16E5 silencing, W12p6 cells were transfected with the E5 siRNA sequence (5′-TGGTATTACTATTGTGGATAA-3′) [25] or the control sequence (5′-AATTCTCCGAACGTGTCACGT-3′) (Qiagen, Valencia, CA, USA), using Lipofectamine 2000 Transfection Reagent (Invitrogen, Carlsbad, CA, USA), according to the manufacturer’s protocol.

## 4. Primers

Oligonucleotide primers necessary for target genes and the housekeeping gene were chosen by utilizing the online tool Primer BLAST [26] and purchased from Invitrogen (Carlsbad, CA, USA). The following primers were used: for FGFR2b target gene, 5′-CGTGGAAAA GAACGGCAGTAAATA-3′ (sense), 5′-GAACTATTTATCCCCGAGTGCTTG-3′ (anti-sense); for FGFR2c target gene, 5′-TGAGGACGCTGGGGAATATACG-3′ (sense), 5′-TAGTCTGGGGAAGCTGTAATCTCCT -3′ (anti-sense); for HPV16 E5 gene, 5′-CGCTGCTTTTGTCTGTGTCT-3′ (sense), 5′-GCGTGCATGTGTATGTATTAAAAA-3′ (antisense); for ESRP1 target gene, 5′-GGCTCGGATGAGAAGGAGTT-3′ (sense), 5′-GCACTTCGTGCAACTGTCC-3′ (antisense); for Snail1 target gene, 5′-GCTGCAGGACTCTAATCCAGA-3′ (sense), 5′-ATCTCCGGAGGTGGGATG-3′ (antisense); for Snail2 target gene, 5′-TGGTTGCTTCAAGGACACAT-3′ (sense), 5′-AGCAAATGCTCTGTTGCAGTG-3′ (antisense); for Zeb1 target gene, 5′-GGGAGGAGCAGTGAAAGAGA-3′ (sense), 5′-TTTCTTGCCCTTCCTTTCTG-3′ (antisense); for Keratin 10 (K10) target gene, 5′-GATGTGAATGTGGAAATGAATGCTG-3′ (sense), 5′-TGTAGTCAGTTCCTTGCTCTTTTCA-3′ (antisense); for Involucrin (Inv) target gene, 5′- CCAGCCTTCTACACCTCAC-3′ (sense), 5′-ACCCATTCCTCCCACTCC-3′ (antisense); for Loricrin (Lor) target gene, 5′-GGTGCTTTGGGCTCTCCTT-3′ (sense), 5′-GAGGTCTTCACGCAGTCCA-3′ (antisense); and for the 18S rRNA housekeeping gene, 5′-AACCAACCCGGTCAGCCCCT-3′ (sense), 5′-TTCGAATGGGTCGTCGCCGC-3′ (antisense). For each primer pair, we performed no-template control and no-reverse-transcriptase control (RT negative) assays, which produced negligible signals.

### 4.1. RNA Extraction and cDNA Synthesis

RNA was extracted using the TRIzol method (Invitrogen) according to the manufacturer’s instructions and eluted with 0.1% diethylpyrocarbonate (DEPC)-treated water. Each sample was treated with DNAase I (Invitrogen) to avoid any possible DNA contamination. Total RNA concentration was quantitated by spectrophotometry. RNA samples were stored at −80 °C. After denaturation in DEPC-treated water at 70 °C for 10 min, 1 µg of total RNA was used for reverse transcription using the iScript^TM^ cDNA synthesis kit (Bio-Rad Laboratoires, Hercules, CA, USA), according to the manufacturer’s instructions. 

### 4.2. PCR Amplification and Real-Time Quantitation

Real-time PCR was performed using the iCycler Real-Time Detection System (iQ5, Bio-Rad Laboratories, Hercules, CA, USA) with optimized PCR conditions. The reaction was carried out in a 96-well plate using iQ SYBR Green Supermix (Bio-Rad Laboratories, Hercules, CA, USA), adding each forward and reverse primer and 1 µL of diluted template cDNA to a final reaction volume of 15 µL. The thermal cycling program was performed as previously described [13]. Real-time quantitation was performed with the help of the iCycler IQ optical system software version 3.0a (Bio-Rad Laboratories, Hercules, CA, USA), according to the manufacturer’s manual. Results are reported as the mean ± standard deviation (SD) from three different experiments in triplicate. 

### 4.3. Statistical Analysis 

The mRNA levels from cytological samples are expressed as the mean ± SD from three independent experiments in triplicate. To model the relationship between the variables, the correlation measures were evaluated by the Pearson test (R^2^) and by linear analysis of the regression curve (CurveExpert data analysis software system). *p* values < 0.05 were assumed to be statistically significant.

## Figures and Tables

**Figure 1 ijms-22-06534-f001:**
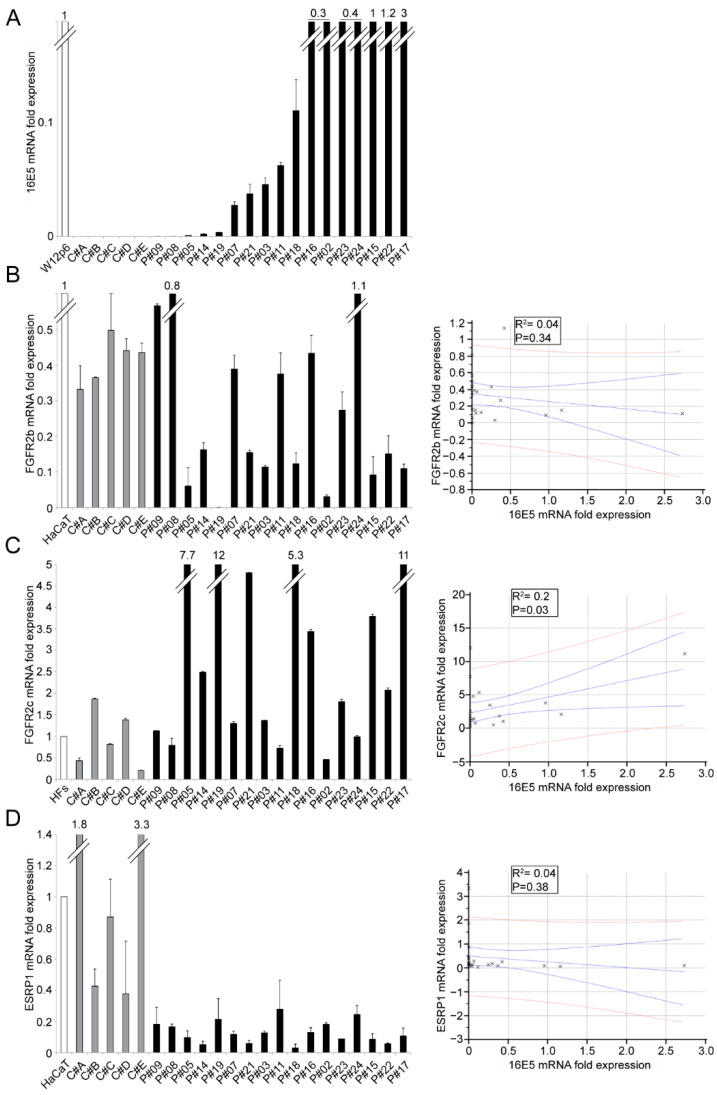
Expression of HPV16E5, FGFR2 isoforms and ESRP1 in LSILs: 16E5 (**A**), FGFR2b (**B**), FGFR2c (**C**) and ESRP1 (**D**) mRNA levels were evaluated by real-time PCR in LSILs and normalized with respect to W12p6 cells, HaCaT cells or HFs, as reported in the graph. Results are expressed as the mean ± SD from three independent experiments in triplicates. The relationships between the variables are expressed in the scatter diagrams in which HPV16E5 expression levels are plotted on the horizontal axis and the FGFR2b, FGFR2c and ESPRP1 expression levels are plotted on the vertical axis. The regression line ±95% confidence limits are shown in blue, and the ±95% prediction bands are in red. R^2^: Pearson’s correlation coefficient value.

**Figure 2 ijms-22-06534-f002:**
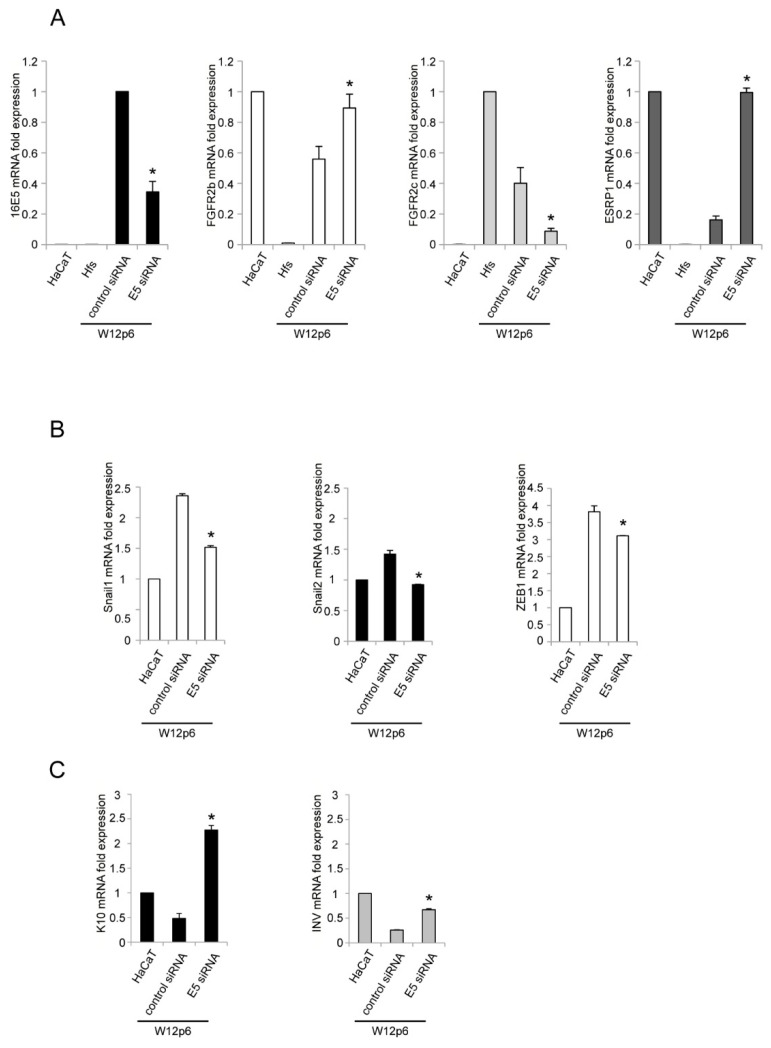
The oncoprotein 16E5 is responsible for ESRP1 downregulation, FGFR2 isoform switching, as well as for the induction of EMT-related transcription factors and the downregulation of keratinocyte differentiation markers. W12p6 cells were transiently transfected with 16E5 siRNA or with unrelated siRNA as a negative control (control siRNA). HaCaT cells and primary cultures of human dermal fibroblasts (Hfs) were used to confirm the tissue-specific expression of FGFR2b and ESRP1 in the epithelial cells, and that of FGFR2c in the mesenchymal cells. After transfection, the 16E5 mRNA levels, as well as those of FGFR2b, FGFR2c, ESRP1 (**A**), Snail1, Snail2 and Zeb1 (**B**), keratin 10 (K10) and involucrin (INV) (**C**) were quantified by real-time PCR. Results are expressed as mean values ± SD. Student’s *t*-test: * *p* < 0.05 vs. the corresponding W12p6 control siRNA cells.

**Figure 3 ijms-22-06534-f003:**
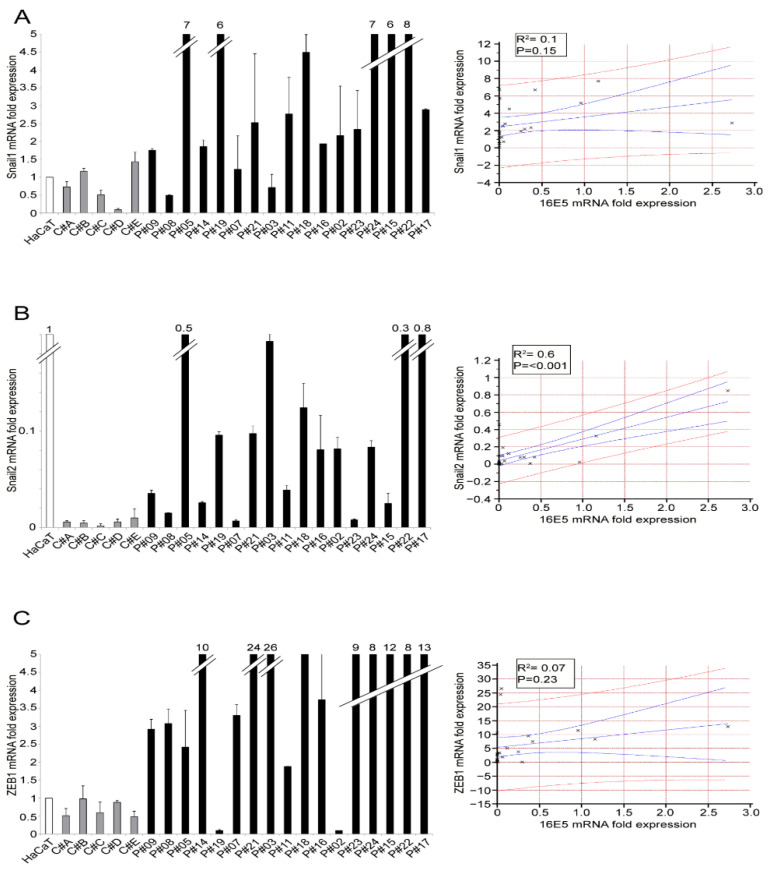
Expression of EMT-related transcription factors in LSILs. Snail1 (**A**), Snail2 (**B**) and Zeb1 (**C**) mRNA levels were evaluated by real-time PCR in LSILs and normalized with respect to HaCaT cells. Results are expressed as mean values ± SD. The relationships between the variables are expressed in the scatter diagrams in which HPV16E5 expression levels are plotted on the horizontal axis and the Snail1, Snail2 and ZEB1 expression levels are plotted on the vertical axis. The regression line ±95% confidence limits are shown in blue, and the ±95% prediction bands are in red. R^2^: Pearson’s correlation coefficient value.

**Figure 4 ijms-22-06534-f004:**
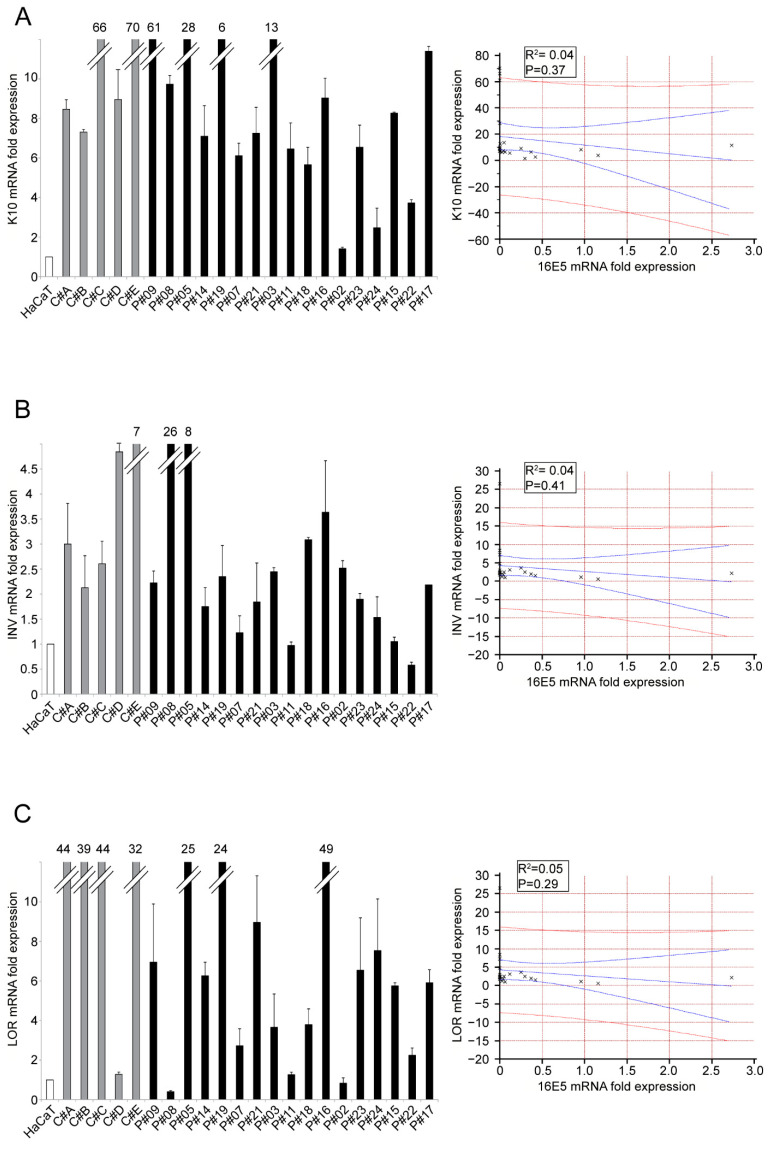
Expression of keratinocyte differentiation markers in LSILs. Keratin 10 (K10) (**A**), involucrin (INV), (**B**) and loricrin (LOR) (**C**) mRNA levels were evaluated by real-time PCR in LSILs and normalized with respect to HaCaT cells. Results are expressed as mean values ± SD. The relationships between the variables are expressed in the scatter diagrams in which HPV16E5 expression levels are plotted on the horizontal axis and the Keratin 10, involucrin and loricrin expression levels are plotted on the vertical axis. The regression line ±95% confidence limits are shown in blue, and the ±95% prediction bands are in red. R^2^: Pearson’s correlation coefficient value.

## Data Availability

The datasets used and/or analyzed during the current study are available from the corresponding author on reasonable request.

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
