# Peer review of "Expression of the E5 Oncoprotein of HPV16 Impacts on the Molecular Profiles of EMT-Related and Differentiation Genes in Ectocervical Low-Grade Lesions"

_ijms, 2021, doi:10.3390/ijms22126534_

Round 1

Reviewer 1 Report

Dear Authors,

congratulations on the results of your work and a manuscript which is very well written, cohesive and understandable as well as very good English is provided in this manuscript. I have only several minor suggestions:

  • Line 31 - I would recommend to mention about other oncogenic HPVs (just to list them as an example). A table comparing oncogenic and non-oncogenic types would also be recommendable
  • Line 42 - please provide the full name of the abbreviation - EMT - before the abbreviation is provided.
  • Please check English once again since I have detected several minor mistakes in the text that would be recommendable to correct before the manuscript could be processed further.

Best wishes with you further research on such important topic.

A Reviewer

Author Response

The comments of the Reviewers have been very helpful, and we have prepared a revised version of our manuscript following their suggestions as follows:

  • As recommended by the Reviewer, the high-risk oncogenic HPV types have been mentioned in the Introduction section and the specific reference has been added in the text (Page:1 row: 39 of the revised manuscript)
  • As suggested, the full name of the abbreviation EMT has been provided in the Introduction section (Page: 2 row: 60 of the revised version).
  • As kindly remarked, minor English mistakes were found in the text which have been corrected.

Reviewer 2 Report

Ranieri D. et al.

It is appreciated the effort that the authors performed in order to show that E5 oncoprotein impacts on the molecular profiles of EMT-related and differentiation genes in ectocervical low-grade lesions. The introduction is well-written and structured. They have data which could possibly support their hypothesis although the most of the presented data do not show any significance. As one figure is missing I could not judge properly the manuscript, although the result part of the figure looks interesting.

Major concerns-comments:

Abstract

I think that in the abstract is not clear what is the most important finding of the manuscript. A clear conclusion could make it possible more obvious.

Results-Discussion

Figure 2 is missing. In the manuscript there are 2 different figure legends one for figure 2 one for 3 but the graphs are the same and they correspond to fig. 3.

It could be crucial for the reader’s understanding to highlight the important findings and the novelty of the manuscript.

Minor concerns-comments:

  • In Figure 1 the graphs should become bigger so you could see clearly especially what is written on the x axis.
  • In line 232 you could say that … was described before (reference number)

Author Response

The comments of the Reviewers have been very helpful, and we have prepared a revised version of our manuscript following their suggestions as follows:

  • We agree with the Reviewer’s remark and, to clarify which are the most important findings of the manuscript, a conclusion has been added in the abstract section, (Page:1 row: 30 of the revised manuscript).
  • We agree with the reviewer: for an inexplicable reason, the figure 2 has been replaced with a duplicate of 3 in the final version developed by the site. In the revised version of the manuscript, we have re-added the right one.
  • As recommended by the reviewer, the importance and the innovative implications of our findings have been further clarified in both the abstract and in the results/discussion sections (Page: 1 row 30; Page:11 row:182 of the revised manuscript).
  • As requested, the graph reported in the left of new Figure 1 has been enlarged.
  • The reference in full in the Materials and Methods section has been replaced by the corresponding reference number (Page: 12 row: 251 of the revised manuscript)

Round 2

Reviewer 2 Report

Dear authors,

thank you very much for the improvement of the manuscript. I think you made clear the goal of your study and you have done very good research.

Minor comment: line 182 you write her but its here

Good luck.

Author Response

  • As recommended by the Reviewer, in page: 9 line: 177 of the revised manuscript, we replaced “her” with “here”.
